# Effects of Physical Activity and Psychological Modification-Based Intervention on Physical Fitness, Physical Activity and Its Related Psychological Variables in Female Adolescents

**DOI:** 10.3390/ijerph18189510

**Published:** 2021-09-09

**Authors:** YoungHo Kim, InKyoung Park

**Affiliations:** Department of Sport Science, Seoul National University of Science and Technology, Seoul 01811, Korea; yk01@seoultech.ac.kr

**Keywords:** female adolescent, psychological strategy, transtheoretical model, physical activity, physical fitness, tabata exercise

## Abstract

Background: The current study investigated the effects of an intervention incorporating physical activity and psychological modification strategy on physical fitness, physical activity levels, and psychological variables related to physical activity in female adolescents. *Methods**:* Sixty female adolescents were recruited from H Middle School. Among them, 30 females (Mage = 14.35 years) were randomly assigned to the experimental group and the rest of 30 to the control group (Mage = 14.47 years) and voluntarily participated in the 12-week intervention. *Results**:* The results indicated that the physical activity stage of female adolescents in the experimental group significantly increased over the 12-week intervention. The results revealed that all of psychological variables in the experimental group significantly improved over the intervention, while participants in the experimental group showed significantly higher scores on most of psychological variables, except cons, than those in the control group after the intervention. Moreover, most of physical fitness components in the experimental group significantly increased over the intervention. *Conclusions**:* The current study confirmed that the physical activity-related psychological intervention was feasible for improving physical and psychological health among female adolescents.

## 1. Introduction

It is broadly documented that regular physical activity is an important health-related behavior to improve adolescents’ physical and psychological health and protect them from various diseases [1]. Also, regular participation in physical activity during adolescence positively affects satisfaction in school and daily life, further preparing them for adulthood and laying the foundation for lifelong physical activity. Despite the various health benefits of regular physical activity, a large number of adolescents around the world have failed to engage in regular physical activity [2,3]. According to a recent survey by the Ministry of Health and Welfare [4], only 32% of Korean adolescents engaged in high-intensity physical activity for at least 3 days a week, and the rate among female adolescents (18%) was found to be significantly lower than among male peers (44.8%). Furthermore, the World Health Organization [3] surveyed adolescents’ physical activity rate and reported that their physical activity rate is very low and might have serious effects on their future health status (i.e., America: male 36%, female 19.5%; England: male 25.3%, female 14.6%; Australia: male 23.2%, female 18.6%). Thus, the adolescent participation rate in physical activity is very low both at home and abroad, which can be a leading cause of deaths due to diabetes and cardiovascular diseases in adulthood as well as negative effects such as an increased likelihood of mental disorders like anxiety and depression [5]. From previous large-scale Korean and international statistics, we should notice that the high rates of physical inactivity in female adolescents are globally reported. Along with the high rates of physical inactivity, adolescents’ physical fitness such as muscle strength, muscle endurance, flexibility, power, and cardiovascular endurance, consistently declined [6]. 

In this regard, it is unclear that why many female adolescents are physically inactive. This primary question can be identified to apply a conceptual framework rooting in the social cognitive theory. Additionally, practical efforts should be directed towards cognitive and behavioral strategies based on psychosocial theories for changing and supporting physical activity initiation and/or maintaining among female adolescents [7]. The transtheoretical model (TTM) is a contemporary psychological paradigm that seeks to understand the adoption and maintenance of health behaviors including physical activity [8]. The TTM indicates that physical activity change happens in a variety of stages over time and that the causes of change involve the cognitive and behavioral processes that individuals participate in at the different stages of change. Moreover, as people progress through the stages of physical activity, they should typically experience increased self-efficacy and pros and decreased cons [9]. 

For more than a decade, many studies across a wide range of populations and settings have shown that there is a significant relationship between physical activity and the TTM constructs [10]. These survey studies can provide valuable findings for the promotion of physical activity among adolescents. Given the significance between psychological attributes and physical activity in adolescents, longitudinal intervention research has focused on increasing physical activity and its linked psychological variables are needed in female adolescents.

In behavioral medicine, recent studies have indicated that the TTM-based comprehensive intervention integrating physical activity and psychological modification strategies is broadly recognized as an effective modality for not only improving physical activity in adolescents, but also in realizing positive changes on the psychological variables linked to physical activity [11,12]. Some studies have indicated that the interventions based on the TTM are effective on promoting physical activity and positively changing psychological variables in various age groups [12,13]. Recently, Jeong and colleagues (2020) applied the intervention combining physical fitness training and cognitive modification strategies for 16 weeks to 120 female adolescents. The results indicated that physical fitness (i.e., power, muscle strength and endurance, flexibility, and cardiovascular endurance), as well as psychological variables such as self-efficacy and perceived benefits, and motivation are significantly improved after the intervention [14]. 

In spite of the critical effects of the intervention in promoting physical activity and positively changing psychological constructs associated with physical activity, most studies have been conducted in western societies; even research combining psychological strategies and physical activity is lacking in South Korea. The purpose of the current study was to investigate the effects of the intervention integrating psychological modification strategy and physical activity on physical activity, physical fitness, and psychological variables in female adolescents.

## 2. Materials and Methods

### 2.1. Participants

Before recruiting the study participant, G*Power analysis was carried out to identify the proper sample size and the magnitude of effect size. From this test, at least 54 participants with an anticipated statistical power of 0.95, a α-error probability of 0.05, and an effect size of 0.25 were required for this study. Based on calculating sample size, a total of 60 students were purposefully recruited from H junior high school, located in the Nowon-gu district, Seoul. Among them, 30 females randomly assigned to the experimental group and the rest of 30 to the control group. The study was conducted according to the guidelines of the Declaration of Helsinki and approved by the institutional review board of Seoul National University of Science and Technology (2020-0565). Table 1 illustrates the characteristics of the participants.

### 2.2. Measures

#### 2.2.1. Physical Fitness

In order to evaluate the participants’ physical fitness, the Physical Activity Promotion System (PAPS) developed by Korean government was applied for the study. The PAPS is a mandatory evaluation program performed once a year on all elementary, middle, and high school students in South Korea. In PAPS, physical fitness consists of five components (i.e., power, muscle strength and endurance, flexibility, cardiovascular endurance) [14]. The current study measured the essential PAPS evaluation items: 50 m run for power, sit-up for muscular strength and endurance), sitting trunk flexion for flexibility, and shuttle run for cardiovascular endurance [15].

#### 2.2.2. Stage of Physical Activity

To measure the participants’ physical activity stages, the stage of change scale for physical activity [16] translated into Korean and used [17]. This questionnaire comprises indicators of five stages for an individual’s level of physical activity (i.e., precontemplation, contemplation, preparation, action, and maintenance). The participants were asked to choose one of the five stages that they thought was consistent with their intention to participate in regular physical activity and their actual physical activity level. According to Kim [17], the reliability of the questionnaire was 0.85, and the internal validity with the LTEQ was 0.81.

#### 2.2.3. Physical Activity Level

In order to measure the amount of physical activity, the leisure time exercise questionnaire (LTEQ) [18] was translated into Korean and applied for this study [19]. In this study, divided into high-intensity, moderate-intensity, and low-intensity categories, the study participants were asked to report how many times during a typical week took part in strenuous (e.g., running, vigorous cycling), moderate (e.g., fast walking, easy swim), and mild (e.g., yoga, golf) physical activity for more than 15 min. The exercise score was multiplied and summed by each intensity level to obtain the total metabolic equivalent (MET) value [(strenuous × 9) + (moderate × 5) + (mild × 3)]. The construct validity of the questionnaire was verified by correlation with the accelerometer (Spearman’s ρ = 0.77), and the reliability was shown by Cronbach’s α = 0.82 [19].

#### 2.2.4. Psychological Variables

The psychometric instruments applied in the current study, were originally developed in English and translated into Korean based on the methodology outlined by Banvile, Desrosiers, and Genet-Volet (2000) [20]. The full translation and validation processes have been described elsewhere [21,22].

To measure the participants’ confidence for doing physical activity, the exercise self-efficacy questionnaire [23] was translated and used to the study [24]. In the questionnaire, the study participants answered their confidence levels to participate in physical activity regularly under the various circumstances on a 5-point Likert scale from” *not confident at all* (1)” to *“very confident* (5)”. The test-retest reliability of the Korean questionnaire was 0.89.

To assess perceived benefits and barriers toward physical activity, the decisional balance scale for Exercise [25] was translated and applied for the study [19]. The scale consists of the two sub-scales (pros and cons with five items each). Study participants indicated their perceived level towards what is described in each question on a 5-point Likert scale ranging from “*not at all important* (1) to “*very much important* (5)”. The test-retest reliability of the Korean questionnaire was 0.91 for pros and 0.89 for cons.

In order to measure participants’ behavioral and cognitive strategies for physical activity, the processes of change scale were translated and applied in the study [26]. By using the 5-point Likert scale, students were asked to answer from “never (1)” to “repeatedly (5)”. In this study, cognitive processes and behavioral processes were influenced by the second-order factors. The five components in cognitive processes are dramatic relief, consciousness-raising, environmental re-evaluation, self-re-evaluation, and social liberation, whereas five components in behavioral are counter conditioning, helping the relationship, self-liberation, reinforcement management, and stimulus control. The internal reliabilities of the Korean version were 0.84 for cognitive processes of change and 0.87 for behavioral processes of change [27].

### 2.3. Intervention

During the 12-week intervention, the study participants in the experimental group were received the Tabata exercise program and psychological modification strategy, while those in the control group read a book freely for 40 min under the supervision of a co-investigator without any experimental treatment. The intervention consists of a Tabata exercise and psychological modification strategy that aims at increasing physical activity and physical fitness and improving psychological constructs over 12 weeks (Table 2). 

The Tabata exercise program used in the study consists of high-intensity interval training (HIIT) regimen which is generally a form of interval training alternating short periods of intense anaerobic exercise with less intense recovery periods, until the participant is too exhausted to continue [28]. In order to motivate the participants to actively engage, a standardized 10-min warm-up consisting of 5 min of slow jogging followed by 5 min of stretching of major muscle groups at first and then the Tabata exercise were performed. The Tabata exercise program was divided into three sessions with a total 14 min (each lasting 4 min). Each session based on the Tabata protocol (20 s work/10 s rest) consisted of 8 cycles of two exercises. Each cycle started with a maximum intensity exercise lasting for 20 s, in which the participant was motivated to perform as many repetitions as possible of a given exercise involving large muscle groups of the entire body which was followed a 10 s-active rest in the form of a low-intensity exercise. The cycles were repeated without any rests between them [29]. After the Tabata training, the final part of the training including flexibility and relaxation exercises was performed for 16 min [30].

The psychological modification strategy was based on the materials applied in previous studies [31,32]. The TTM constructs such as self-efficacy, pros, cons, and cognitive and behavioral processes of change, were applied for the motivational and behavioral modification strategies to promote physical activity and physical fitness and positively change psychological variable in female adolescents. At the initial stage, the intervention was aimed at raising accurate awareness of the significance of physical activity and physical fitness. The intermediate stage was emphasized to apply practical techniques to enhancing self-efficacy and social support associated with regular physical activity. The final stage of the intervention was focused on stressing cognitive and behavioral reinforcement techniques for the participants to sustain physical fitness and to promote physical activity participation [13,32].

### 2.4. Data Analysis

Descriptive data (e.g., mean, standard deviation, kurtosis, and skewness) were analyzed to demonstrate the demographic characteristics of the study participants. A McNemar chi-square (χ^2^) test was performed to test differences in the distribution of the physical activity stages between the experimental group and the control group over 12-week intervention. Two-way repeated-measure ANOVA was conducted to examine differences in physical activity, physical fitness, and psychological constructs between the experimental group and control group over the intervention. All statistical analyses were performed using SPSS Win 26.0 (IBM Corp., Armonk, NY, USA).

## 3. Results

### 3.1. Changes in the Stage of Physical Activity

Table 3 shows the stage transitions of study participants in both experimental and control groups over the 12-week intervention. There is no difference in the stage distributions of physical activity between the experimental and control groups at baseline. In the experimental group, the physical activity stage of participants significantly increased after the intervention; however, there was no statistical significance in the control group. A McNemar chi-square (χ^2^) test was conducted to examine the differences in stage of change over the 12 weeks and revealed significant differences between baseline and 12 weeks (McNemar χ^2^ values = 79.62, *df* = 7, *p* < 0.001).

### 3.2. Changes in Physical Fitness and Physical Activity

Two-way repeated-measures ANOVA was performed to investigate the effects of the intervention on physical fitness and physical activity of participants. According to Table 4, most of physical fitness components in the experimental group significantly increased over the intervention [*F(1,58)* = 55.58 for power, *p* < 0.001; *F(1,58)* = 20.87 for cardiovascular endurance, *p* < 0.001; *F(1,58)* = 15.82 for muscle strength and endurance, *p* < 0.01; and *F(1,58)* = 12.23 for BMI, *p* < 0.01]. In addition, the experimental group showed significantly higher records on several physical fitness components than the control group after the intervention [*F(1,58)* = 50.25 for power, *p* < 0.001; *F(1,58)* = 20.17 for cardiovascular endurance, *p* < 0.001; and *F(1,58)* = 6.64 for muscle strength and endurance, *p* < 0.05]. For physical activity (METs), only participants in the experimental group showed significant increase ([*F(1,58)* = 12.23, *p* < 0.01).

### 3.3. Changes in Psychological Variables

Table 5 shows the result of two-way repeated-measures ANOVA to examine the effects of the intervention on psychological variables of participants. The results indicated that participants in the experimental group displayed significant increase in pros [*F(1,58)* = 51.47, *p* < 0.001], self-efficacy [*F(1,58)* = 49.31, *p* < 0.001], cognitive processes [*F(1,58)* = 17.17, *p* < 0.01], behavioral processes [*F(1,58)* = 12.32, *p* < 0.01], and cons [*F(1,58)* = 3.77, *p* < 0.05] over the intervention. Additionally, participants in the experimental group showed significantly higher scores on most of psychological variables, except cons, than those in the control group after the intervention [*F(1,58)* = 79.45 for self-efficacy, *p* < 0.001; *F(1,58)* = 63.13 for pros, *p* < 0.001; *F(1,58)* = 29.97 for behavioral processes, *p* < 0.001; and *F(1,58)* = 17.17 for cognitive processes, *p* < 0.01].

## 4. Discussion

The current study aimed at identifying the effects of the intervention integrating Tabata exercise and psychological modification strategy on physical activity, physical fitness, and psychological variables in female adolescents. The findings indicated that the stages of physical activity of participants in the experimental group were significantly changed over the intervention, and this was supported by previous findings [33,34]. Meaningfully, all participants (100%) in the experimental group were in the precontemplation, contemplation, and preparation stages at baseline, but 76.7% of them reported being in the action and maintenance stage over the 12-week intervention.

In addition, the findings indicated that the study participants in the experimental group also showed increases in their physical activity levels over the intervention. It might be a possible reason to be considered that the participants in this study might be willing to engage in the intervention or they satisfied to exercise with their friends because the study participants did not exercise at all or irregularly exercised before. However, such interpretation needs to be considered with caution, because this finding has been obtained from only Korean female adolescents.

The result found that the study participants’ physical fitness significantly different between the experimental and control group after the 12-week intervention. In addition, physical fitness in the experimental group showed significantly increases over the intervention. These findings are supported by previous studies, demonstrating that high-intensity interval training increased cardiovascular endurance with maximal oxygen intake among female adolescents [35,36]. Furthermore, some studies indicated that the Tabata exercise was effective to improve muscle strength and endurance and power among female students [35,36]. However, the current finding revealed that there was no significant difference in flexibility between the experimental and control group over the intervention. It is plausible to explain that the Tabata protocols applied in the current study were not included an exercise to improve flexibility; instead, flexibility might have increased naturally given the nature of adolescence as a period of rapid physical development, rather than the effect of the exercise intervention.

Furthermore, the current finding indicated that the study participants as a whole experienced increases in their self-efficacy, pros, and the use of the cognitive and behavioral processes of change over the intervention, with decreases in their cons for being physically activity. This is important to document because many physical activity intervention studies have failed to document changes in the theoretical variables that served as the basis of the physical activity intervention in the first place [30,31]. A large number of studies widely documented that self-efficacy, cognitive and behavioral processes, and decisional balance (i.e., pros and cons) are the significant psychological determinants of physical activity in various cross-sectional studies [21,32]. Considering the lack of research on psychological strategies integrating physical activity (Tabata exercise) and psychological strategy in behavioral medicine, this study implies that the current intervention modality is a significant method to promote physical activity and positively change psychological constructs related to physical activity. Kang and Lee applied a psychological factor-based physical activity program to the physical education class of junior high school and reported that this intervention was significant to promote physical activity participation and its related competence and enjoyment [37]. More recently, Kwan and the colleagues conducted systematic review and meta-analysis to identify the effect of psychological modification strategy on improving physical activity [38]. This study concluded that the strategy is effective at increasing the time spent on physical activity and energy expenditure in physical activity, using cognitive and behavioral strategies, and improving perceived benefits and self-efficacy. In a broad viewpoint, furthermore, it needs to discuss about the associations between psychological variables and physical activity with the different people of different societies. Recent studies carried out in Malaysia and China indicated that individuals’ psychological constructs such as beliefs, values, intention, and confidence play a critical role to improve health-related behaviors including physical activity and therefore the mechanism of linkage between psychological variables and a broad range of health behaviors should be understood in a socio-cultural perspective [39,40].

There are a number of limitations should be considered. First, the study participants may have spent time reading other physical activity-related posts outside of the study, which we were unable to objectively assess and account for in analyses. Second, given the relatively small and homogeneous sample, findings may have limited generalizability to the diverse population. Third, although the measures used in this study were psychometrically sound, they were all self-report measures and therefore item interpretation, recall, and social desirability were not controlled for. 

## 5. Conclusions

The current study is the significant attempt to employ the comprehensive approach combining physical activity and psychological strategy in female adolescents. The current study confirmed that the physical activity-related psychological intervention was feasible for improving physical and psychological health among female adolescents. Therefore, further research should continue to examine new ways to capitalize on the psychology-focused physical activity strategy and functionalities that promote physical activity involvement and its adherence for other age groups.

## Figures and Tables

**Table 1 ijerph-18-09510-t001:** Study participants’ characteristics at baseline.

Variables	Experimental Group (*n* = 30)	Control Group (*n* = 30)	*t*	*p*
Age (yr)	14.35 ± 0.73	14.47 ± 0.68	0.27	0.59
Height (cm)	160.75 ± 5.38	158.55 ± 5.84	1.51	0.14
Weight (kg)	52.79 ± 9.27	52.08 ± 7.77	0.32	0.75
BMI (kg/m²)	20.39 ± 3.2	20.71 ± 2.88	0.41	0.68

BMI = body mass index, *t* = t-value, *p* = practical significance.

**Table 2 ijerph-18-09510-t002:** Physical activity and psychological modification-based intervention.

Week	Physical Activity (One a Week) Tabata Exercise	Psychological Modification Strategy (2 Times/Week)	
		Topic	Strategy
1	Explanation of the effect of Tabata exercise Explanation and demonstration of how to implement Tabata exercise	Understanding of physical activity Goal setting	Introduction of psychological strategy Awareness of physical inactivity-related problems Setting the target physical activity amount
2	Warm-up (10 min) Streching (6 min) Tabata exercise (14 min) 3 sessions/4 min. per session	Identification of current level	Physical activity test (Mets score) Physical fitness test Self-efficacy test Perceived benefits and barriers test
3		Identification of one’s problems Setting of new plan for behavioral change	Finding one’s reasons for not doing physical activity Discussing one’s lifestyle Setting an achievable physical activity plan
4		Physical activity	Understanding the importance and effect of physical activity Understanding the significance of physical fitness training Discussion one’s physical activity habits
5–6		Self-efficacy	Understanding the meaning and scope of physical activity Identifying the relationship between self-efficacy and physical activity Enhancing confidence for promoting physical activity
7–8		Motivation Perceived benefits and barriers	Understanding the importance of motivation and perception toward physical activity Recognizing the benefits and barriers of physical activity Exploring the way to enhance motivation and perceived benefits
9		Social support	Understanding the meaning and resources of social sup-port Asking support from friends and families
10–11		Awareness change	Sharing one’s thought and experiences about obesity and physical activity with others Replacing negative factors for physical activity with positive ones
12		Setting new goal for physical activity	Identifying the attainment of one’s goal and rewarding it Testing physical activity, physical fitness and psychological factors Setting new physical activity goal for maintaining active lifestyle

**Table 3 ijerph-18-09510-t003:** Changes in the stage distributions of physical activity over the intervention.

	Stages of Physical Activity					
	PC	CO	PR	AC	MA	Total ^a^
Baseline-12 weeks						
PC	1(0)	1(3)	1(0)	0(0)	0(0)	3(3)
CO	0(5)	1(9)	3(1)	6(0)	0(0)	10(15)
PR	0(6)	0(4)	0(2)	16(0)	1(0)	17(12)
AC	0(0)	0(0)	0(0)	0(0)	0(0)	0(0)
MA	0(0)	0(0)	0(0)	0(0)	0(0)	0(0)
Total ^b^	1(11)	2(16)	4(3)	22(0)	1(0)	30(30)

*Note.* Parentheses are the control group. ^a^ Frequencies indicate the sum of stage of change distribution at baseline. ^b^ Frequencies indicate the sum of stage of change distribution at 12-week time period. PC: precontemplation; CO: contemplation; PR: preparation; AC: action; MA: maintenance.

**Table 4 ijerph-18-09510-t004:** Changes in physical fitness and physical activity over the intervention.

Variables	Group	N	Baseline (M ± SD)	Week 12 (M ± SD)	Group *F(1,58)*	Time *F(1,58)*	Wilks Lambda(λ)
Cardiovascular endurance	EG	30	42.30 ± 5.96	50.23 ± 7.22	20.17 ***	20.87 ***	0.33
	CG	30	39.93 ± 5.38	41.10 ± 5.39			
Flexibility	EG	30	15.67 ± 8.87	16.99 ± 7.78	0.17	1.59	0.04
	CG	30	15.55 ± 8.89	16.23 ± 8.85			
Muscular strength and Endurance	EG	30	24.89 ± 4.61	26.78 ± 4.54	6.64 *	15.82 **	0.43
	CG	30	23.62 ± 2.89	24.01 ± 2.72			
Power	EG	30	138.30 ± 19.38	165.76 ± 16.54	50.25 ***	55.58 ***	0.51
	CG	30	132.33 ± 13.62	134.13 ± 13.35			
Physical activity (METs)	EG	30	24.73 ± 19.02	36.83 ± 16.93	7.04 **	12.23 **	0.37
	CG	30	24.03 ± 15.65	26.43 ± 15.84			

* *p* < 0.5, ** *p* < 0.01, *** *p* < 0.001. EG: experimental group, CG: control group. BMI; body mass index. METs: metabolic equivalents.

**Table 5 ijerph-18-09510-t005:** Changes in psychological variables over the intervention.

Variables	Group	N	Baseline (M ± SD)	Week 12 (M ± SD)	Group *F(1,58)*	Time *F(1,58)*	Wilks Lambda(λ)
Pros	EG	30	2.90 ± 0.32	3.97 ± 0.46	63.14 ***	51.47 ***	0.51
	CG	30	2.74 ± 0.58	3.00 ± 0.37			
Cons	EG	30	2.48 ± 0.56	1.83 ± 0.31	1.33	3.77 *	0.08
	CG	30	2.40 ± 0.82	2.54 ± 0.41			
Self-efficacy	EG	30	2.87 ± 0.56	3.76 ± 0.24	79.45 ***	49.31 ***	0.62
	CG	30	2.56 ± 0.72	2.23 ± 0.37			
Cognitive processes	EG	30	2.88 ± 0.71	3.60 ± 0.52	17.17 **	16.60 **	0.25
	CG	30	2.81 ± 0.90	2.87 ± 0.41			
Behavioral processes	EG	30	2.68 ± 0.65	3.22 ± 0.35	29.97 ***	12.32 **	0.30
	CG	30	2.74 ± 0.82	2.70 ± 0.60			

* *p* < 0.05, ** *p* < 0.01, *** *p* < 0.001. EG: experimental group, CG: control group.

## Data Availability

The data included in the present study are available upon request from corresponding author.

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
