# Peer review of "Effects of Physical Activity and Psychological Modification-Based Intervention on Physical Fitness, Physical Activity and Its Related Psychological Variables in Female Adolescents"

_ijerph, 2021, doi:10.3390/ijerph18189510_

Round 1

Reviewer 1 Report

The manuscript is interesting, methodologically well designed, and the results support the discussion. However, I have the following comments.

I. Major Comments:
1. The introduction does not present relevant background information on the benefits of physical activity (eg disease prevention).

2. Improve the legends of all tables. The legend of each table should guide whoever reads the manuscript and facilitate understanding.

3. In the discussion I suggest to include (briefly) aspects related to the benefits that would be generated. Especially for women's health.

4. Considering the age of the people studied, it is possible to project results in older people ?, which occurs in aging. Discuss this point.

II. Minor comments:
1. Improve the writing of the objective of the study.
2. I suggest including a brief paragraph on the projections of the study, considering the results obtained.

Author Response

First of all, the authors are very appreciate for valuable comments suggested by the reviewers. Additionally, we must express warm appreciation to the editor-in-chief to give a chance to improve the manuscript. We have done our best to follow the reviewers’ suggestions. Most of the comments have been fixed with adding some extra sentences in red throughout the entire manuscript and several comments have been explained for better understanding.

Reviewer 1

  1. Major Comments:
    1. The introduction does not present relevant background information on the benefits of physical activity (eg disease prevention).

-I have corrected.

  1. Improve the legends of all tables. The legend of each table should guide whoever reads the manuscript and facilitate understanding.

-I have checked and fixed.

  1. In the discussion I suggest to include (briefly) aspects related to the benefits that would be generated. Especially for women's health.

-I have added.

  1. Considering the age of the people studied, it is possible to project results in older people ? which occurs in aging. Discuss this point.

-I have done

Reviewer 2 Report

Generally speaking, the author want to explore the effect of physical exercise and psychological modification strategy on female adolescents’ physical activity, physical fitness and several psychological variables related to physical activity by experimental design. It can be evidently seen that some important and meaningful results can be found from the present study, but some errors and potential faults should be carefully and seriously revised and corrected, and the specific details can be seen as follows. Thank you !

  1. It can be seen that the two authors have been from the same institution, so it is unnecessary to mark the numerical symbol “1”or “2”in the upper right corner of the name, and the author’s address can be retained only one.

  1. In the section of abstract, the results should contain some expressions about physical fitness changes before and after 12 weeks intervention. In addition, the conclusion should be reconstructed, due to it seems more likely implications or suggestions.

  1. The ”physical fitness”should be added to the keyword.

  1. In table 1, the symbol “t”and “P” should be displayed by italics.

  1. The secondary title 2.2 “MEASURES”should be changed into 2.2 “Measures”.

  1. From Line 121 to 138, these contents should be classified to the evaluation of physical activity instead of psychological variables, so the author should add a three-level title, namely physical activity, and put these contents inside this title.

  1. In the section of 2.3 intervention, some errors have been revised and several potential biases should attach much importance for authors afterward. At first, the authors says that warm-up contains 10 Min, but 5 Min for slow jogging and 6 Min for stretching of major muscles are equal to 11 Min. It can be seen that this is an evident error. Secondly, the total time for the control group are 40 Min, so for the experiment group, except for 10 Min with the warm-up and 14 Min with tabata exercise, the rest for the flexibility and relaxation should be 16 Min rather than for several minutes. This expression is so unscrupulous and inaccurate. Thirdly, how to ensure participants from the experiment group can reach to the high-intensity level during exercise, and in this article no any objective indicators could support and verify this. The so-called description, “until the participant is too exhausted to continue”, is only a subjective judgment. In case they do not try their best to perform this exercise program, and this subjective evaluation may bot be reliable and valid. At last, the author might not tell us that during the experiment, whether all participants may not do any other exercise programs, because this will affect the final experiment results, and even the accuracy of conclusion to this study.

By the way, from the perspective of the accuracy and reliability for casual relation, this study should set four groups, namely one control group and three experiment groups including performing tabata exercise, psychological modification strategy, combing tabata exercise and psychological modification strategy, respectively. In the present study, the author only sets two groups, so it is a pity that we can not get more convincing and persuasive casual conclusion. This is a reminder for authors.

  1. In the section of 2.4 data analysis, number 2 should be located in the upper right corner of “χ”, instead of the right of this symbol.

  1. Line 218, values in the bracket should be deleted.

  1. In table 3, the“stages of physical activity”should be in the middle position rather than the left in this table.

  1. In the section of 3.3 “changes in the psychological variables”, the five motivations of physical activity can not be found in Table. 5, and only two psychological variables, namely cognitive processes and behavioral processes have been displayed in this table. That means these five physical activity motivations are equal to cognitive and behavioral processes ? But, according to the description of BREQ-2, no any evidence can be certify this. So, please check it seriously.

  1. The references are not listed according tothe corresponding standards and requirementsof the IJERPH. For example, the year should be used by bold font, the journal name displayed by abbreviation, and except for the first letter, some journals’ title are all used the capital letter such as reference 11, 15, 16, 32, 39. So, this section must be carefully and responsibly revised by authors.

Author Response

First of all, the authors are very appreciate for valuable comments suggested by the reviewers. Additionally, we must express warm appreciation to the editor-in-chief to give a chance to improve the manuscript. We have done our best to follow the reviewers’ suggestions. Most of the comments have been fixed with adding some extra sentences in red throughout the entire manuscript and several comments have been explained for better understanding.

Reviewer 2

Generally speaking, the author want to explore the effect of physical exercise and psychological modification strategy on female adolescents’ physical activity, physical fitness and several psychological variables related to physical activity by experimental design. It can be evidently seen that some important and meaningful results can be found from the present study, but some errors and potential faults should be carefully and seriously revised and corrected, and the specific details can be seen as follows. Thank you !

  1. It can be seen that the two authors have been from the same institution, so it is unnecessary to mark the numerical symbol “1”or “2”in the upper right corner of the name, and the author’s address can be retained only one.

-I have corrected.

  1. In the section of abstract, the results should contain some expressions about physical fitness changes before and after 12 weeks intervention. In addition, the conclusion should be reconstructed, due to it seems more likely implications or suggestions.

- I have added some sentence and fixed

  1. The ”physical fitness”should be added to the keyword.

- I have added

  1. In table 1, the symbol “t”and “P” should be displayed by italics.

- I have fixed

  1. The secondary title 2.2 “MEASURES”should be changed into 2.2 “Measures”.

- I have fixed

  1. From Line 121 to 138, these contents should be classified to the evaluation of physical activity instead of psychological variables, so the author should add a three-level title, namely physical activity, and put these contents inside this title.

 - I have fixed

  1. In the section of 2.3 intervention, some errors have been revised and several potential biases should attach much importance for authors afterward. At first, the authors says that warm-up contains 10 Min, but 5 Min for slow jogging and 6 Min for stretching of major muscles are equal to 11 Min. It can be seen that this is an evident error. Secondly, the total time for the control group are 40 Min, so for the experiment group, except for 10 Min with the warm-up and 14 Min with tabata exercise, the rest for the flexibility and relaxation should be 16 Min rather than for several minutes. This expression is so unscrupulous and inaccurate. Thirdly, how to ensure participants from the experiment group can reach to the high-intensity level during exercise, and in this article no any objective indicators could support and verify this. The so-called description, “until the participant is too exhausted to continue”, is only a subjective judgment. In case they do not try their best to perform this exercise program, and this subjective evaluation may bot be reliable and valid. At last, the author might not tell us that during the experiment, whether all participants may not do any other exercise programs, because this will affect the final experiment results, and even the accuracy of conclusion to this study.

By the way, from the perspective of the accuracy and reliability for casual relation, this study should set four groups, namely one control group and three experiment groups including performing tabata exercise, psychological modification strategy, combing tabata exercise and psychological modification strategy, respectively. In the present study, the author only sets two groups, so it is a pity that we can not get more convincing and persuasive casual conclusion. This is a reminder for authors.

-Many thanks for valuable comments on keeping reliability and validity of experimental design. I totally agree with this. Unfortunately, we checked how much participants reach to strenuous activity depending on their subjective expression. Additionally, we missed to classify the group into 4. We will keep in mind this point and further study will very carefully aware this. Thanks again.

  1. In the section of 2.4 data analysis, number 2 should be located in the upper right corner of “χ”, instead of the right of this symbol.

 - I have fixed

  1. Line 218, values in the bracket should be deleted.

 - I have fixed

  1. In table 3, the“stages of physical activity”should be in the middle position rather than the left in this table.

 - I have fixed

  1. In the section of 3.3 “changes in the psychological variables”, the five motivations of physical activity can not be found in Table. 5, and only two psychological variables, namely cognitive processes and behavioral processes have been displayed in this table. That means these five physical activity motivations are equal to cognitive and behavioral processes ? But, according to the description of BREQ-2, no any evidence can be certify this. So, please check it seriously.

-It is totally terrible mistake. Motivation is not psychological variable which was dealt in this study. We have deleted the BREQ-2 in Measures and added processes of change there. Thanks for your point.

  1. The references are not listed according tothe corresponding standards and requirementsof the IJERPH. For example, the year should be used by bold font, the journal name displayed by abbreviation, and except for the first letter, some journals’ title are all used the capital letter such as reference 11, 15, 16, 32, 39. So, this section must be carefully and responsibly revised by authors.

-I have fixed

Reviewer 3 Report

Dear Authors,

Thanks for giving me the chance to read this manuscript, “Effects of Physical Activity and Psychological Modification-Based Intervention on Physical Fitness, Physical Activity and Its related Psychological Variables in Female Adolescents”. The current study tries investigate the effects of an intervention incorporating physical activity and psychological modification strategy on physical fitness, physical activity levels, and psychological variables related to physical activity in female adolescents.

Generally, it is an interesting topic. However, there are several points worth noting which must be appropriately addressed before further processing.

  1. Method and sample recruitment

The article mentioned the sample strategy that 60 students were purposefully recruited from H junior high school. However, we did not know how participants were selected. Convenience sample? Structured sampling? Detailed information was needed to help readers to get a better understanding of your experimental setting.

  1. Discussion on countries’ difference

The authors are advised to use makes more contribution to this study. For example, authors could further discuss the different sample of different countries in some psychological variables, such as self-efficacy (Qin et al., 2019). Thus, it could help readers to have a more precise scope of this study.

Ref:

Qin, Z.; Song, Y.; Jin, Y. Green worship: The effects of devotional and behavioral factors on adopting electronic incense products in religious practices. Int. J. Environ. Res. Public Health 2019, 16.

  1. Measure issue

In the section 2.2.2, it is advised for authors to have a table to summarize all the measures and protocol used in this study.

  1. Analysis issue

In this study, authors used Two-way repeated-measures ANOVA to perform the analysis. However, the results only showed the main effects of these variables. It is indeed interesting to see the interaction effect of time and group which would shed light on an in-depth explanation.

  1. Language issue

There are many typos and inappropriate language issue in the current version. Authors are advised to use a professional proofreading service to improve the quality of language.

To sum up, I personally like this paper and its contributions, and it is definitely a potentially publishable paper. However, the issues mentioned above should be carefully addressed in order to further proceed. Hope these suggestions help.

Author Response

First of all, the authors are very appreciate for valuable comments suggested by the reviewers. Additionally, we must express warm appreciation to the editor-in-chief to give a chance to improve the manuscript. We have done our best to follow the reviewers’ suggestions. Most of the comments have been fixed with adding some extra sentences in red throughout the entire manuscript and several comments have been explained for better understanding.

Reviewer 3

Thanks for giving me the chance to read this manuscript, “Effects of Physical Activity and Psychological Modification-Based Intervention on Physical Fitness, Physical Activity and Its related Psychological Variables in Female Adolescents”. The current study tries investigate the effects of an intervention incorporating physical activity and psychological modification strategy on physical fitness, physical activity levels, and psychological variables related to physical activity in female adolescents.

Generally, it is an interesting topic. However, there are several points worth noting which must be appropriately addressed before further processing.

  1. Method and sample recruitment

The article mentioned the sample strategy that 60 students were purposefully recruited from H junior high school. However, we did not know how participants were selected. Convenience sample? Structured sampling? Detailed information was needed to help readers to get a better understanding of your experimental setting.

 -The participants were purposefully recruited and it is described. Thanks

  1. Discussion on countries’ difference

The authors are advised to use makes more contribution to this study. For example, authors could further discuss the different sample of different countries in some psychological variables, such as self-efficacy (Qin et al., 2019). Thus, it could help readers to have a more precise scope of this study.

-Thanks. I have revised the discussion section. 

  1. Measure issue

In the section 2.2.2, it is advised for authors to have a table to summarize all the measures and protocol used in this study.

 -Thanks. Because this study has already 5 table, I am afraid the number of table if it adds one more table. Instead, I used new sub-heading to improve readability. Please accept this change.

  1. Analysis issue

In this study, authors used Two-way repeated-measures ANOVA to perform the analysis. However, the results only showed the main effects of these variables. It is indeed interesting to see the interaction effect of time and group which would shed light on an in-depth explanation.

 -Thanks. The interaction effect was not significant and so the authors did not mention. Please accept this.

  1. Language issue

There are many typos and inappropriate language issue in the current version. Authors are advised to use a professional proofreading service to improve the quality of language.

To sum up, I personally like this paper and its contributions, and it is definitely a potentially publishable paper. However, the issues mentioned above should be carefully addressed in order to further proceed. Hope these suggestions help.

Round 2

Reviewer 1 Report

Authors made all changes suggested. Manuscript can be accepted in present form. 

Author Response

I am very appreciate for your thoughtful comments.

Accordingly, I have carefully done all spelling check.

Reviewer 3 Report

Although the authors have addressed most of my concerns, I did not see an improvement in the discussion part. For example, authors could further discuss the different samples of different countries in some psychological variables, such as self-efficacy (Qin et al., 2019). Thus, it could help readers to have a more precise scope of this study.

See this paper

Qin, Z., Song, Y., & Jin, Y. (2019). Green worship: The effects of devotional and behavioral factors on adopting electronic incense products in religious practices. International journal of environmental research and public health16(19), 3618.

Author Response

Dear Reviewer

I am very appreciate for your thoughtful comment which is valuable to improve implication of the manuscript. Ultimately, it should be helpful for international readers to consider their own study using a physical activity internvetion.

As you suggestion, I have added the following sentences in the Discussion section with citing the reference that you recommended. Thanks.

"In a broad viewpoint, furthermore, it needs to discuss about the associations between psychological variables and physical activity with the different people of different societies. Recent studies carried out in Malaysia and China indicated that individual‘s psychological constructs such as beliefs, values, intention, and confidence play a critical role to improve health-related behaviors including physical activity and therefore the mechanism of linkage between psychological variables and a broad range health behaviors should be understood in a socio-cultural perspective [40,41]".

  • Qin, Z.; Song, Y.; Jin, Y. Green worship: The effects of devotional and behavioral factors on adopting electronic incense products in religious practices. Int. J. Environ. Res. Public Health. 2019, 16(19), 3618. https://doi.org/10.3390/ijerph16193618
  •